# Effects of Lithium Ions on tPA-Induced Hemorrhagic Transformation under Stroke

**DOI:** 10.3390/biomedicines12061325

**Published:** 2024-06-14

**Authors:** Valentina A. Babenko, Elmira I. Yakupova, Irina B. Pevzner, Alexey D. Bocharnikov, Ljubava D. Zorova, Kseniya S. Fedulova, Oleg A. Grebenchikov, Artem N. Kuzovlev, Andrey V. Grechko, Denis N. Silachev, Parvaneh Rahimi-Moghaddam, Egor Y. Plotnikov

**Affiliations:** 1A.N. Belozersky Research Institute of Physico-Chemical Biology, Lomonosov Moscow State University, 119991 Moscow, Russia; nucleus-90@yandex.ru (V.A.B.); elmira.yaku@gmail.com (E.I.Y.); irinapevzner@mail.ru (I.B.P.); balex5000@yandex.ru (A.D.B.); lju_2003@list.ru (L.D.Z.); xenia.fedulova@yandex.ru (K.S.F.); silachevdn@genebee.msu.ru (D.N.S.); 2V.I. Kulakov National Medical Research Center of Obstetrics, Gynecology and Perinatology, 117997 Moscow, Russia; 3Advanced Engineering School “Intelligent Theranostics Systems”, Sechenov First Moscow State Medical University, 119992 Moscow, Russia; 4Federal Research and Clinical Center of Intensive Care Medicine and Rehabilitology, 107031 Moscow, Russia; ogrebenchikov@fnkcrr.ru (O.A.G.); artem_kuzovlev@fnkcrr.ru (A.N.K.); avgrechko@fnkcrr.ru (A.V.G.); 5Department of Pharmacology, School of Medicine, Iran University of Medical Sciences, Tehran 14496-14535, Iran; rahimi.p@iums.ac.ir

**Keywords:** tissue plasminogen activator, hemorrhagic transformation, photothrombosis, oxygen-glucose deprivation, lithium salts

## Abstract

Thrombolytic therapy with the tissue plasminogen activator (tPA) is a therapeutic option for acute ischemic stroke. However, this approach is subject to several limitations, particularly the increased risk of hemorrhagic transformation (HT). Lithium salts show neuroprotective effects in stroke, but their effects on HT mechanisms are still unknown. In our study, we use the models of photothrombosis (PT)-induced brain ischemia and oxygen-glucose deprivation (OGD) to investigate the effect of Li^+^ on tPA-induced changes in brain and endothelial cell cultures. We found that tPA did not affect lesion volume or exacerbate neurological deficits but disrupted the blood–brain barrier. We demonstrate that poststroke treatment with Li^+^ improves neurological status and increases blood–brain barrier integrity after thrombolytic therapy. Under conditions of OGD, tPA treatment increased MMP-2/9 levels in endothelial cells, and preincubation with LiCl abolished this MMP activation. Moreover, we observed the effect of Li^+^ on glycolysis in tPA-treated endothelial cells, which we hypothesized to have an effect on MMP expression.

## 1. Introduction

Stroke is one of the leading causes of death and disability worldwide [1]. Currently, there are two main treatment options for acute ischemic stroke. One is the intravenous administration of tissue plasminogen activator (tPA) within 4.5 h of symptom onset [2]. However, this thrombolytic therapy has limitations such as hemorrhagic transformation, neurotoxicity, and a short treatment window [3,4]. An alternative therapy is a mechanical thrombectomy, but it has its limitations as well [5,6,7,8]. One of them is that many ischemic strokes are caused by occlusion of cerebral arteries that cannot be reached by intra-arterial catheters [9]. Therefore, it is still appropriate to modify thrombolytic tPA therapy and deepen our understanding of the mechanisms underlying its adverse effects in order to minimize them.

One of the mechanisms of injury associated with thrombolytic therapy with tPA involves matrix metalloproteinases (MMPs) [10]. MMPs are known to be upregulated in response to acute central nervous system injury, including ischemia and trauma [11,12]. Treatment with tPA in acute ischemic stroke has been reported to result in significantly increased levels of pro-MMP-9 and activated MMP-9 [4,10,11]. These proteolytic enzymes, which belong to the extracellular protease family, are capable of degrading various components of the extracellular matrix. In the context of hemorrhagic transformation (HT), uncontrolled MMP activity after tPA-mediated reperfusion is responsible for the degradation of important cerebrovascular proteins such as collagen, laminin, and zonula occludens-1 (ZO-1) [13,14,15]. This degradation can compromise the integrity of capillary walls, leading to vascular leakage and rupture [11]. While the source of MMP-9 production remains a subject of ongoing debate [16], endothelial cells are thought to make a possible contribution by enhancing the secretion of MMP-9 and MMP-2 after tPA treatment [17].

Lithium salts possess pronounced neuroprotective effects in acute cerebral pathologies [18,19]. Randomized clinical trials on the effect of Li^+^ in post-stroke patients showed improved motor recovery after early treatment with a low dose of lithium carbonate [20]. A protective effect of lithium ions was observed for endothelial cells, astrocytes [21], and neurons [22] exposed to ischemia. In both oxygen-glucose deprivation (OGD) and a mice stroke model, Li^+^ treatment resulted in an increase in the expression of tight junction proteins [23], indicating stabilization of the blood–brain barrier (BBB), which was associated with the inhibition of MMP-9 activity [23]. Chronic Li^+^ treatment for 14 days resulted in neurovascular remodeling via an increase in MMP-9 levels [24]. These observations suggest that the lithium impact on MMP activity is multifaceted. However, the comprehensive effects of lithium treatment in combination with thrombolytic tPA therapy remain unexplored.

In this study, we aimed to elucidate the mechanisms underlying tPA-induced HT, with particular attention to the role of MMPs, and to evaluate the effects of concomitant treatment with lithium chloride in a rat model of stroke induced by photothrombosis (PT) and in endothelial cell cultures subjected to OGD in vitro.

## 2. Materials and Methods

### 2.1. Experimental Animals

Experiments were performed on outbred rats maintained on a 12/12 h light/dark cycle and at a temperature of 22 ± 2 °C. The rats were used according to the protocols evaluated and approved by the Animal Ethics Committee of the A.N. Belozersky Research Institute of Physico-Chemical Biology (Protocol number 2/20, dated 12 February 2020). All methodologies were aligned with the guidelines prescribed by the Federation of Laboratory Animal Science Associations (FELASA). The experimental animals were divided into four groups: a group without photothrombosis (PT) and pharmacological intervention (Control, n = 10); a group exposed to PT of the prefrontal cortex followed by saline administration (PT, n = 15); a group exposed to PT followed by administration of tPA (PT + tPA, n = 13); and a group exposed to PT followed by LiCl treatment and then receiving tPA (PT + tPA + LiCl, n = 8).

### 2.2. Photothrombosis Model

The Rose Bengal PT model offers several methodological advantages, including consistent localization and size of cerebral infarcts, keeps mortality rates low, and requires relatively simple surgical protocols. The model was selected due to its inability to produce red thrombi, consistent with our specific goal of exploring the side effects of tPA in ischemic brain tissue. The established protocol for photothrombosis has been described in previous studies [25,26]. For the procedure, rats were anesthetized by intraperitoneal injection with 6% chloral hydrate (300 mg/kg). Rose bengal (Sigma-Aldrich, 198250, Burlington, MA, USA), dissolved in sterile saline, was then infused into the jugular vein at a concentration of 3% (40 mg/kg). Then, the intact skull surface was irradiated with a 520-nm laser (∼100 mW, 3-mm beam diameter) for 15 min, using a stereotactic frame to precisely target 0.5 mm posterior and 2.5 mm lateral to Bregma. Rats in the PT + tPA + LiCl group received a bolus of 180 mg/kg LiCl intraperitoneally (Sigma-Aldrich, Burlington, MA, USA) 1 h after PT. For tPA treatment, the PT + tPA and PT + tPA + LiCl groups received an intravenous bolus injection of tPA at a dose of 10 mg/kg in 1 mL (Boehringer Ingelheim, Ingelheim-am-Rhein, Germany) under brief isoflurane anesthesia 6 h after stroke modeling. In the PT control group, rats received 1 mL of an isotonic saline solution at the same time after PT.

### 2.3. Limb-Placing Test

The neurological status of rats was evaluated on the 3rd and 6th days post-photothrombosis using a limb-placing test (LPT). The protocol based on the protocol by De Ryck [27] modified by Jolkkonen [28] was used. For each task, the following scores were used: 2 points, normal response; 1 point, delayed and/or incomplete response; 0 points, no response. The total score for seven tasks was evaluated with a maximum possible score of 14, indicative of the absence of neurological deficit.

### 2.4. Assessing Asymmetry of Forelimb Use in a Test Cylinder

The assessment of forelimb use asymmetry was conducted using a cylinder test, which measures the spontaneous exploration behaviors of the animal against the walls of a transparent cylinder (dimensions: 30 cm in height and 20 cm in diameter). During the test, rats were observed for 5–8 min while inside the cylinder, and their activities were recorded with a camcorder positioned opposite the mirror under the cylinder. This setup allowed for the quantification of independent uses of the contralateral (affected) and ipsilateral (unaffected) forelimbs, as well as instances where both forelimbs were used simultaneously, specifically while the rat was in a rearing posture engaging with the cylinder walls. The frequency of forelimb use was determined using the formula: 100 × (contr + ½ × simult)/(ipsi + simult + contr), where ‘contr’ represents the usage of the contralateral forelimb, ‘ipsi’ the usage of the ipsilateral forelimb, and ‘simult’ the simultaneous use of both forelimbs [29].

### 2.5. Evans Blue Extravasation

Brain Evans blue dye (EBD) leakage was analyzed on the 7th day after the PT according to a previously described protocol [30] with modifications. Briefly, rats were anesthetized with chloral hydrate and subsequently received an i.v. injection of 2% Evans Blue (MP Biomedicals, LLC, Illkirch-Graffenstaden, France) in saline solution (4 mL/kg) into the jugular vein. After 30 min, the animals underwent intracardiac perfusion with 200 mL of phosphate-buffered saline (PBS) to remove the intravascular Evans Blue. The brain was then isolated and photographed, and the injured area was removed and weighed. The same piece of the contralateral hemisphere was excised and also weighed.

Brain samples were homogenized in saline (90 mg of brain per 150 µL of saline). Then 50% trichloroacetic acid in 50% ethanol was added to homogenate in a 3:1 ratio, followed by centrifugation at 10,000× *g* for 20 min. The supernatant was collected, and the fluorescence of EBD was measured using a microplate reader (ZENYTH 3100, Anthos Labtec Instruments GmbH, Wals, Austria) at an excitation wavelength of 535 nm and an emission wavelength of 625 nm. Relative extravasation of EBD in brain tissue was calculated using external standards (50–1000 ng/mL) dissolved in the same solvent.

### 2.6. MRI Scan Evaluation of Brain Damage

Brain damage in animals was investigated seven days post-stroke using magnetic resonance imaging (MRI) with a Bio-Spec 70/30 system (Bruker, Billerica, MA, USA) equipped with a 7 T magnetic field and a gradient system of 105 mT/m. T2-weighted images were acquired employing a RARE (Rapid Acquisition with Relaxation Enhancement) spin echo pulse sequence with parameters set to TR = 6000 ms, TE = 63.9 ms, slice thickness of 0.8 mm, a field of view of 4.2 × 3.1 cm, and a matrix size of 256 × 384, yielding a resolution of 0.164 × 0.164 mm/pixel. The total duration of imaging was approximately 4 min and 48 s. For the duration of the MRI procedure, animals were anesthetized with isoflurane and positioned within a thermostatically controlled holder.

### 2.7. Oxygen-Glucose Deprivation in Cell Culture

We used endothelial cell line EA.hy926 provided by Prof. C. J. Edgell (Carolina University, USA) derived from human endothelium and represented the phenotype of microvascular endothelial cells [31]. Cells were cultured in DMEM/F-12 (1:1) medium, with 10% fetal bovine serum (FBS, BioSera, Cholet, France), penicillin and streptomycin (100 U/mL), 2 mM L-glutamine (PanEco, Moscow, Russia), and HT (PanEco, Moscow, Russia). Cell cultures were incubated at 37 °C in 5% CO_2_, and the medium was replaced every 3 days. For modeling of ischemic stroke, in vitro cells were exposed to oxygen-glucose deprivation (OGD). EA.hy926 was seeded onto 96-well plastic plates (Corning, Lowell, MA, USA) with 25,000 cells/well in the culture medium and cultured 2 days before OGD. The cell medium was replaced with Dulbecco′s Phosphate Buffered Saline (8 mM NaHPO_4_, 137.4 mM NaCl, 1.47 mM KH_2_PO_4_, 2.7 mM KCl, 0.9 mM CaCl_2_, 0.5 mM MgCl_2_) (BioinnLabs, Rostov-on-Don, Russia). Then cells were incubated in a hypoxic atmosphere with 1% O_2_ and 99% N2 at 37 °C for 4 h. Control cells were cultured under normoxic conditions. The number of replicates for each group was 3.

### 2.8. Cell Viability Assay

Cell viability was assessed by 3-(4,5-Dimethyl-2-thiazolyl)-2,5-diphenyl-2H-tetrazolium bromide (MTT) assay. The cells were incubated with 5 mg/mL MTT for 1 h at 37 °C. Then the medium with MTT was replaced by 100 μL of DMSO to dissolve the accumulated viable cells in formazan. Absorbance was measured at λ = 595 nm on the microplate reader spectrophotometer (ZENYTH 3100, Anthos (Biochrom), Holliston, MA, USA). The number of replicates for each group was 10.

### 2.9. Drug Treatment

EA.hy926 cells were incubated for 3 h with 1 mM LiCl (Sigma-Aldrich, Burlington, MA, USA) before OGD and the cell viability assay. Immediately after OGD, cells were exposed to reoxygenation by replacing saline with the culture medium with different concentrations (0.4, 0.8, 1.6, and 3.2 μM) of tPA for 22 h. For Western blot and zymography, cell cultures were incubated with 1.6 μM tPA or with 1.6 μM tPA and 1 mM LiCl for 22 h. For the metabolic assay, cells were incubated with 1.6 μM tPA for 22 h. All drugs were diluted in a standard cell medium (composition described above).

### 2.10. Western Blot Analysis

The EA.hy926 was seeded in 75sm2 culture flasks (Corning, USA) and cultured under standard conditions until a monolayer was obtained within a week. After drug treatment, total cell lysates were prepared using the RIPA buffer (Merk Millipore, Darmstadt, Germany).

Cell lysates were mixed with sample buffer containing 0.125 M Tris-HCl (pH 6.8), 4% sodium dodecyl sulfate, 40% glycerol, 0.05% bromophenol blue, and 10% 2-mercaptoethanol, boiled for 5 min, and used for immunoblotting. Samples were loaded on 5–20% Tris-glycine polyacrylamide gels (10–15 µg of total protein per line). After electrophoresis, gels were blotted onto polyvinylidene difluoride (PVDF) membranes (Amersham Pharmacia Biotech, Newcastle, UK). Membranes were blocked with 5% skim milk (Serva, 42590.01, Heidelberg, Germany) in PBS containing 0.05% Tween-20, and subsequently incubated with primary antibodies: anti-MMP-2 1:1000 rabbit (Abcam, Cambridge, UK), anti-MMP-9 1:1000 rabbit (Abcam, UK), anti-GAPDH 1:2000 mouse (HyTest, Moscow, Russia), and anti-b-actin 1:2000 mouse (Sigma, USA). Then, the membranes were incubated with secondary antibodies—anti-rabbit IgG or anti-mouse IgG conjugated with horseradish peroxidase 1:5000 (IMTEK, Moscow, Russia)—and detected using the Advansta Western Bright ECL kit (Advansta, San Jose, CA, USA) using the Chemidoc™ MP Imaging System (Bio-Rad, Hercules, CA, USA). The protein concentration was measured using bicinchoninic acid-based assay (Sigma-Aldrich, Burlington, MA, USA). The number of replicates for each group was 3.

### 2.11. MMP Zymograghy

Cell lysates were mixed with non-reducing sample buffer containing 0.125 M Tris-HCl (pH 6.8), 4% sodium dodecyl sulfate, 40% glycerol, and 0.05% bromophenol blue and loaded without boiling on 10% Tris-glycine polyacrylamide gel, containing 0.2% of gelatin (10–15 µg of total protein per line). The Collagenase II sample was used as a positive control. After electrophoresis, the gel was incubated twice for 15 min in 25 mL of Novex Zymogramm renaturing buffer (Invitrogen, Carlsbad, CA, USA), and incubated for 15 min in 25 mL of freshly prepared Novex Zymogramm developing buffer (Invitrogen, USA). Then the gel was incubated in 25 mL of developing buffer for 16 h at 36 °C. Finally, the gel was stained with Coomassie Brilliant Blue solution for 30 min at 95 °C. After destaining, gelatinolytic activities of MMPs were detected as clear bands against the blue background of Coomassie Brilliant Blue stained gelatin. MMP activity was evaluated using Image Lab software, version 6.1 (Bio-Rad, Hercules, CA, USA). The number of replicates for each group was 3.

### 2.12. Analysis of Glycolysis and Respiration

EA.hy926 was seeded in Seahorse 8-well miniplates at 15,000 cells/well and cultured for 2 days under standard conditions. The cell medium was replaced with the assay medium (XF DMEM, 1 mM sodium pyruvate) after two washings. The oxygen consumption rate (OCR) and extracellular acidification rate (ECAR) were measured using the Seahorse XFp analyzer (Seahorse Biosciences, Billerica, MA, USA). Three measurements were performed in basal conditions and four measurements were taken after injections of each compound affecting bioenergetics: D-glucose (10 mM, Sigma-Aldrich, Burlington, MA, USA), oligomycin (4.5 μM, Sigma-Aldrich, Burlington, MA, USA), carbonyl cyanide m-chlorophenyl hydrazone (CCCP) (10 μM, Sigma-Aldrich, Burlington, MA, USA), and rotenone and antimycin (2.5 and 4 μM, respectively, Sigma-Aldrich, Burlington, MA, USA). The total measurement time was 130 min. Data analysis was performed using XFp Wave software, version 2.6.1 (Seahorse Biosciences, Billerica, MA, USA). The number of replicates for each group was 3.

### 2.13. Statistical Analysis

Statistical analysis was performed with GraphPad Prizm 7 (GraphPad Software Inc., Boston, MA, USA). Data are presented as mean ± standard error of the mean (SEM). Data were tested for normal distribution using the Shapiro–Wilk test. Comparisons between multiple groups in the in vivo experiment (LPT and EBD data) and in the in vitro experiment (EA.hy926 cell survival and zymography data) were performed using a two-way ANOVA with repeated measures followed by a post-hoc Sidak test, and other comparisons between multiple groups were performed using a one-way ANOVA followed by a post-hoc Sidak test. The significance level was set at *p* ≤ 0.05 (*—*p* ≤ 0.05, **—*p* ≤ 0.005, ***—*p* ≤ 0.001).

## 3. Results

### 3.1. The Effect of Concurrent Thrombolytic Therapy with LiCl on Brain Damage

PT caused damage to the sensorimotor cortex indicated as a hyperintensive-signal area in MR images (Figure 1). PT caused local damage to the brain covering approximately 46.8 mm^3^ infarct volumes (Figure 1A). In the group treated with tissue tPA subsequent to PT induction, there was a trend observed toward an increased lesion volume of 55.9 mm^3^; however, this increase was not statistically significant (*p* = 0.35, one-way ANOVA). The concurrent therapy with LiCl only demonstrated a trend for recovery from PT + tPA exposure to the formation of brain lesions with a volume of 44.1 mm^3^ (*p* = 0.27, one-way ANOVA).

In the present study, we used the limb-placing and cylinder tests to evaluate the neurological deficits (Figure 2). Before the start of the experiment, control animals did not show any neurological dysfunction and their neurological status was scored by 14 points in the limb-placing test. Using the limb-placing test, we found sensorimotor dysfunction of the contralateral fore- and hindlimbs after the PT (Figure 2A). On the third day after PT, the sum of the LPT scores were 6.3 ± 0.6, 5.2 ± 0.3, and 8.7 ± 1.6 in the animals in PT, PT + tPA, and PT + tPA + LiCl groups, respectively. On the sixth day post-photothrombosis, a statistically significant reduction in neurological deficits was observed in the PT + tPA + LiCl group (9.5 ± 2.3) compared to the PT (6 ± 0.6) group and PT + tPA (5.3 ± 2.3) group (*p* ≤ 0.05 and *p* ≤ 0.05, respectively, two-way ANOVA). On the sixth day, the difference between PT + tPA and PT + tPA + LiCl was significant in neurological status and was 2-fold higher during combined therapy (*p* ≤ 0.005, two-way ANOVA).

Using the cylinder test, we found that PT results in asymmetry in the use of the forelimbs. Normally, rats use both the left and right limbs in the same proportion when exploring the walls of the closed space in the glass cylinder test (Figure 2B). Seven days following the stroke simulation, the usage frequency of the contralateral paw decreased to 42.9%. Post-injection of tPA further diminished this frequency to 39.7%. Concurrent treatment with lithium chloride partially ameliorated the decrease in limb usage, bringing the frequency to 45.2%. Nonetheless, these differences were not statistically significant.

### 3.2. Evaluation of the Blood–Brain Barrier Integrity Using EBD

BBB integrity was assessed using the EBD leakage assay (Figure 3). Rats subjected to PT exhibited no significant EBD extravasation into the brain parenchyma, evidenced by blue staining in the ischemic and peri-infarct regions with a mean intensity of 34.3 ± 16.3 a.u. (Figure 3A). Conversely, tPA treatment led to an extensive increase in EBD staining (119.5 ± 46.42 a.u.), which was significant compared with the contra hemisphere (*p* ≤ 0.005, two-way ANOVA) and the damaged area in the PT group (*p* ≤ 0.005, two-way ANOVA). The brains of rats with LiCl treatment showed significantly lower EBD content in the infarct area (30.7 ± 7.2 a.u.) compared to the PT + tPA group (*p* ≤ 0.005, two-way ANOVA).

### 3.3. Effects of tPA on OGD-Induced Death of Endothelial Cells

The tPA had a cytotoxic effect on EA.hy926 cells both alone and in combination with OGD. In the range of tPA concentrations (0.4, 0.8, 1.6, and 3.2 µM), a concentration-dependent decrease in the viability of EA.hy926 cells was observed (Figure 4A). More pronounced cell death was observed during the incubation of endothelial cells with tPA after exposure to 4-h OGD, indicating a summation of the negative effects of tPA and ischemia (Figure 4A).

Pre-incubation with 1 mM LiCl had no significant effect on the cell viability of EA.hy926 in control groups and after 22-h treatment with 0.8 and 1.6 μM tPA in normoxia (Figure 4B). A similar effect was observed after 4-h OGD where lithium ions did not increase endothelial cell survival (Figure 4C).

### 3.4. Effects of LiCl on tPA-Induced MMPs Increase

Western blot analysis revealed significantly increased levels of MMP-2 in EA.hy926 cells after 22-h incubation with 1.6 μM tPA (Figure 5A, *p* ≤ 0.05). A similar trend toward the upregulation of MMP-9 levels was observed in cells treated with tPA (Figure 5B). Incubation with LiCl resulted in a drop in both MMP-2 and MMP-9 levels compared to incubation with tPA cells alone (Figure 5).

Zymographic analysis also demonstrated an increase in both pro-MMP-2 and active MMP-2 isoforms after incubation of EA.hy926 with tPA (Figure 5C). LiCl treatment caused a decrease in both MMP-2 forms (Figure 5C).

### 3.5. Energetic Metabolism Analysis

For the assessment of cell bioenergetics, OCR and ECAR were measured. Treatment of EA.hy926 cells with tPA at a dose of 1.6 µM had no significant effect on OCR compared to non-treated cells. We observed conventional responses to the addition of D-glucose, uncoupler CCCP, and respiratory chain inhibitors (oligomycin, rotenon + antimycin A). However, tPA treatment did not result in any significant changes in OCR (Figure 6A). Similarly, lithium chloride treatment failed to influence endothelial cell respiration.

Glycolytic activity measured as ECAR (L-lactate production) was significantly affected by tPA. Treatment with 1.6 μM tPA for 22 h resulted in elevated ECAR in EA.hy926 cells (Figure 6C). The most significant changes were observed in basal glycolytic levels after tPA incubation (Figure 6D). At the same time, we showed an increase in the intracellular level of GAPDH after tPA treatment of endotheliocytes, which also indicates the activation of glycolysis (Figure 6E). However, LiCl treatment, when combined with tPA, had no effect on the ECAR and glycolytic parameters of these cells (Figure 6D). Nevertheless, in endotheliocytes incubated with LiCl along with tPA, there was a decrease in GAPDH levels (Figure 6F). Thus, tPA increased the glycolytic activity of endotheliocytes, while LiCl did not abolish these effects, although it reduced the amount of one of the glycolysis enzymes.

## 4. Discussion

PT is a minimally invasive and reproducible model of ischemic stroke simulating most of the pathological cascades in the brain. The method is based on photostimulation of the Rose Bengal leading to vascular endothelium damage, platelet aggregation, and clot formation [25,26]. It has been supposed that the resistance of the PT model to fibrinolytic therapy is thought to be related to the formation of a platelet-rich but fibrin-poor clot as a result of the photochemical reaction [32]. So, we used this model for the study of the negative effects of tPA thrombolytic therapy of stroke and its modulation by Li^+^. Therefore, photothrombosis is ideal for our research on the potential side effects of tPA in ischemic brain tissue. Our evaluation of tPA alone and in conjunction with LiCl revealed no significant impact on lesion size based on MRI assessments (Figure 1). We did not expect the lesion to be reduced by the use of LiCl, as it was administered to the animals when the lesion was almost formed, and it was difficult to reduce its size. tPA, in turn, did not cause a significant increase in lesion size, although the tendency to increase size was observed, as we had expected. However, the results of a functional assessment of a neurological deficit with LPT on the 6th day after PT demonstrated a neuroprotective effect of lithium in tPA thrombolytic therapy (Figure 2A). Lithium is a well-known neuroprotector that protects all key cells of the neurovascular unit, including endotheliocytes [21]. However, its effect has never been tested for its ability to prevent hemorrhagic transformation caused by tPA. The Evans Blue extravasation technique revealed a notable increase in BBB permeability following tPA administration, leading to a marked enhancement in hemorrhagic transformation. Conversely, pretreatment with lithium for one hour post-PT demonstrated a reduction in EBD extravasation within the lesion and peri-infarct zone, (Figure 3). Haupt et al. (2020) have shown that in mouse middle cerebral artery model occlusion (MCAO), lithium (2 mmol/kg, each at 24 h and at 48 h after MCAO) stabilizes the post-stroke blood–brain barrier via MAPK/ERK/pathway activation and decreases the activity and expression of MMP-9 independent of caveolin-1 [23]. Also, they demonstrate the ability of lithium (final concentration of 1.25 mM) to prevent oxygen-glucose-deprived endothelial cells from tight junction loss [23].

For further investigation of the mechanism of Li^+^ effects on tPA adverse effects, we used the in vitro OGD model on endothelial EA.hy926 cells. It has been demonstrated that an increase in the tPA concentration from 0 to 3.2 µM results in a gradual decrease in cell survival under both normoxic and OGD conditions, indicating a direct cytotoxic effect of tPA (Figure 4A). tPA has already been shown to affect cells (e.g., neurons, astrocytes) in a plasminogen-dependent and independent fashion and cause neurotoxicity [33,34]. Investigation of tPA toxicity for neurons, astrocytes, and brain endothelial cells demonstrated that exposure to tPA induced neuronal death at  ≥2 µM and astrocyte and endothelial cells at  ≥5 μM. OGD exacerbated tPA toxicity, affecting these cells as low as 0.1 µM [35]. It should be noted that OGD activates MMP-9 release from astrocytes and endothelial cells, but not neurons, and that OGD with tPA increases MMP-9 activation [35]. The exact mechanism by which tPA enhances ischemic injury remains to be further investigated. Possible mechanisms supposed Sonic hedgehog pathway involvement in the tPA-induced reduction in trans-endothelial electrical resistance in brain microvascular endothelial cells [36]. Inhibition of this signaling might contribute to the tPA-induced disruption of the endothelial barrier and the promotion of cell injury [36].

Interestingly, our results show that Li^+^ has no effect on cell viability in the presence of tPA, both under normoxic and OGD conditions (Figure 4B,C). We investigated the possible influence of lithium on MMP activity since it is known that MMPs, especially MMP-9 and MMP-2, play a crucial role in hemorrhagic transformation [35,37,38]. The proteolytic activity of these enzymes contributes to the destruction of the BBB [39], and the effect of tPA on MMPs is apparently one of the main reasons for the increased risk of hemorrhagic transformation during thrombolytic therapy [10]. Our findings indicate that tPA exposure elevates MMP-2 and MMP-9 levels in EA.hy926 cells, a response that can be mitigated through prior incubation with LiCl (Figure 5A,B). Our results suggest that the effect of Li^+^ on MMPs in endothelial cells may contribute to the maintenance of BBB integrity after treatment without compromising the therapeutic efficacy of tPA.

At the moment, the mechanism by which LiCl reduces MMP-9 activity is not yet fully understood. It has been suggested that the reduction in MMP-9 activity caused by lithium may be connected to the stimulation of the MAPK/ERK-1/2 pathway, as lithium has been shown to increase the phosphorylation of the pathway’s components. The inhibition of the MAPK/ERK1/2 pathway reversed the Li^+^-induced decline in MMP-9 activity and promoted BBB permeability [23]. The role of MAPK/ERK1/2 signal pathways in MMP-9 expression is well-known [39,40,41] including those in tPA-induced MMP-9 activation [17]. An additional pathway is Wnt/β-catenin signaling upregulation after Li^+^ treatment, which saves tight junctions and endothelial function and is accompanied by decreased MMP-9 expression [42]. However, in some conditions, this signaling results in the stimulation of MMP-9 expression [43,44,45], which requires the clarification of the mechanism of Li^+^ effects on BBB disruption.

One of the most intriguing findings in our study is the impact of tPA on the bioenergetic parameters of endothelial cells. Our results revealed a notable increase in glycolysis following tPA treatment, coupled with elevated levels of GAPDH expression. Although the connection between MMP activity and glycolysis in endothelial cells or the neurovascular unit has not been extensively studied, there is existing evidence suggesting that MMPs can be regulated through glycolysis in other cell types. For example, suppressing glycolysis has been shown to downregulate MMP9 (but not MMP2) in human epithelial carcinoma cells [46] and macrophages in vitro. For instance, in some cancers, the expression of MMP-9 has been reported to be regulated by NF-kB [47]. However, exposure of cancer cells to the glycolysis inhibitor switched the regulation of MMP-9 to the SIRT-1-dependent mechanism instead of the NF-kB signaling pathway [46]. Conversely, promoting aerobic glycolysis through tankyrase activation has been linked to increased MMP-9 expression via the Wnt/β-catenin signaling pathway in ovarian cell lines [48]. The precise mechanistic details of this glycolysis-MMP relationship remain unclear, though it is plausible that HIF-1-dependent regulation plays a role. Treatment by tPA was shown to increase HIF-1 expression in the ischemic brain [49]. Given that HIF-1 orchestrates the transcription of genes involved in glycolysis as well as MMPs, it is plausible to suggest that HIF-1 may serve as a crucial mediator in the link between MMP activity and the glycolytic shift observed in tPA-treated cells.

Limitations of this study. The use of a single cell line, namely EA.hy926 endothelial cells, does not provide a complete picture of the processes occurring in the brain at the level of the entire neurovascular unit. Since tPA and lithium can also act on astrocytes and neurons, we plan to investigate the effect of tPA on these cells in the future. Furthermore, we have not analyzed the long-term effects of tPA therapy, which requires new experiments, as in our case, the analysis of many parameters required the sacrificing of an animal.

## 5. Conclusions

Our study demonstrates that lithium chloride treatment has the potential to mitigate cerebral hemorrhage, BBB disruption, and neurological deficits induced by tPA therapy in ischemic stroke. We found that LiCl effectively counteracts the tPA-induced elevation of MMP-2/9 levels, which may provide insight into the mechanisms underlying BBB preservation. The protective effects of lithium ions may be mediated by decreasing glycolysis in endothelial cells, which can affect MMP levels. In addition to our findings, we have uncovered a complex interplay between GAPDH expression and its modulation by tPA, subsequently impacting glycolysis. This intricate relationship highlights the potential role of lithium in modulating this process. The vascular protective effects of lithium ions may be mediated not only through the downregulation of glycolysis in endothelial cells but also through the consequential impact on MMP levels. The ability of lithium to potentially influence both glycolysis and MMP expression underscores its promising therapeutic potential in preserving vascular integrity and ameliorating the adverse effects associated with tPA treatment in ischemic stroke. Further investigations into the precise mechanisms underlying these interactions could provide valuable insights for developing novel therapeutic strategies for ischemic stroke management.

## Figures and Tables

**Figure 1 biomedicines-12-01325-f001:**
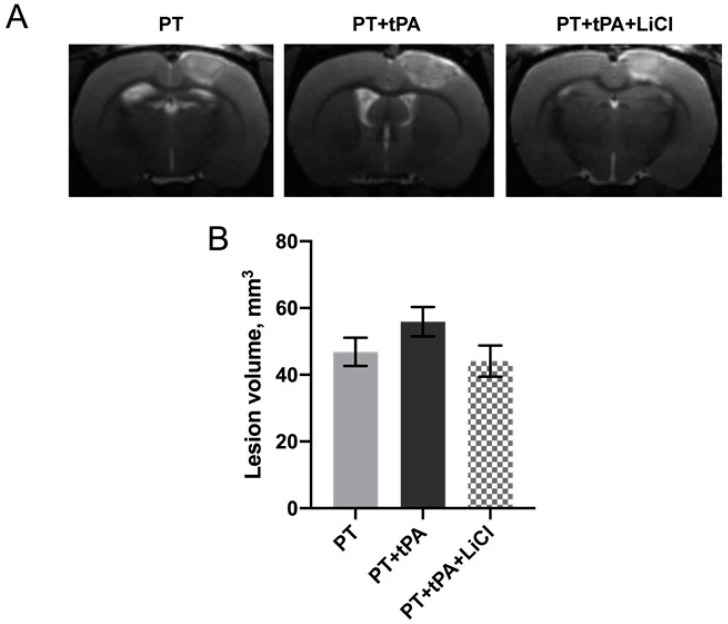
Evaluation of the neuroprotective effect of concurrent thrombolytic therapy with LiCl on the severity of brain damage on the 7th day after PT in rats. (**A**) Representative brain section obtained by T2-weighed MRI (image covered the center of ischemic area). Hyperintensive regions refer to ischemic areas. (**B**) Volume of brain lesions caused by PT in the PT evaluated by using MRI with analysis of T2-weighted images. No significant difference between the groups was found (one-way ANOVA test). The data are presented as mean ± SEM.

**Figure 2 biomedicines-12-01325-f002:**
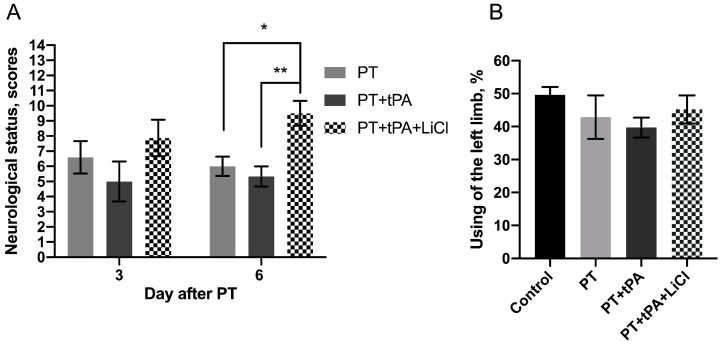
(**A**) Neurological status scores estimated in the limb-placing test (3rd and 6th days after the PT). The data are presented as mean ± SEM. *—*p* ≤ 0.05 in PT vs. PT + tPA + LiCl, **—*p* ≤ 0.005 in PT + tPA vs. PT + tPA + LiCl group on day 6 (two-way ANOVA test). (**B**) PT-induced limb asymmetry measured in the cylinder test on the 7th day after the surgery. The data are presented as mean ± SEM.

**Figure 3 biomedicines-12-01325-f003:**
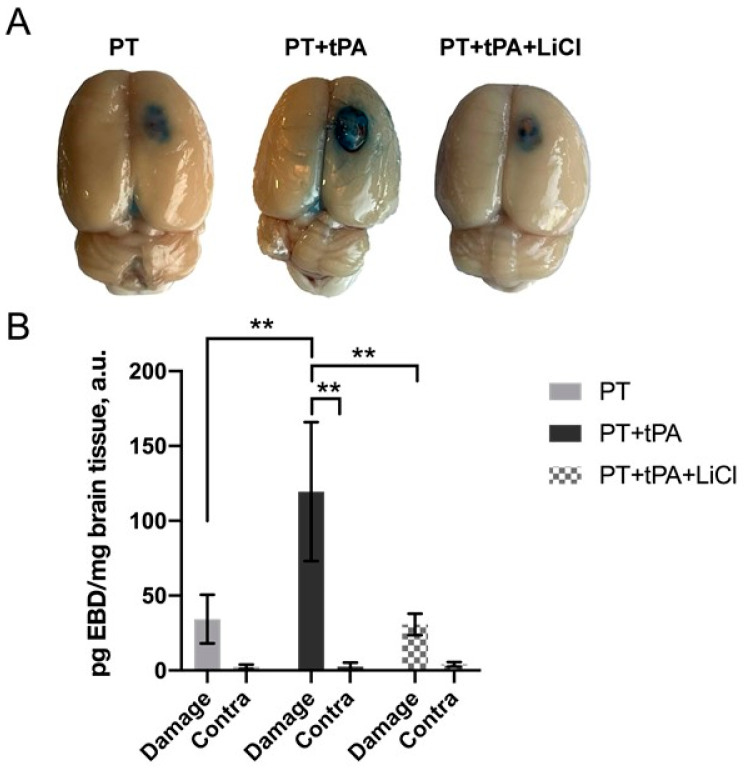
Representative images of the brain with EBD in the PT area in the right hemisphere (**A**). (**B**) The levels of EBD accumulation in brain tissue of damaged and contralateral hemisphere. **—*p* ≤ 0.005 (two-way ANOVA test).

**Figure 4 biomedicines-12-01325-f004:**
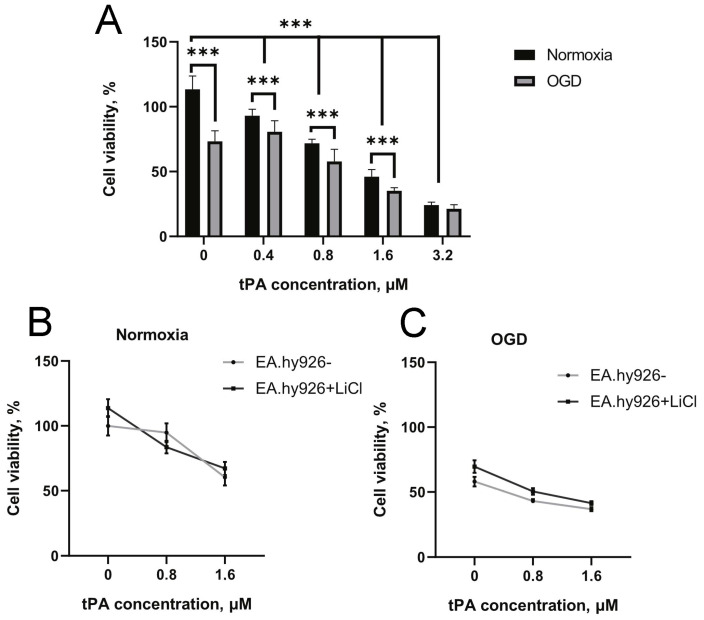
Survival of EA.hy926 cells exposed to increasing concentrations of tPA. (**A**) Toxicity of tPA in normoxia. Effect of LiCl pretreatment on tPA-induced cell death in normoxia (**B**) or after OGD (**C**). The data are presented as mean ± SEM ***—*p* ≤ 0.001 (two-way ANOVA test). Viability of control cells was taken as 100%.

**Figure 5 biomedicines-12-01325-f005:**
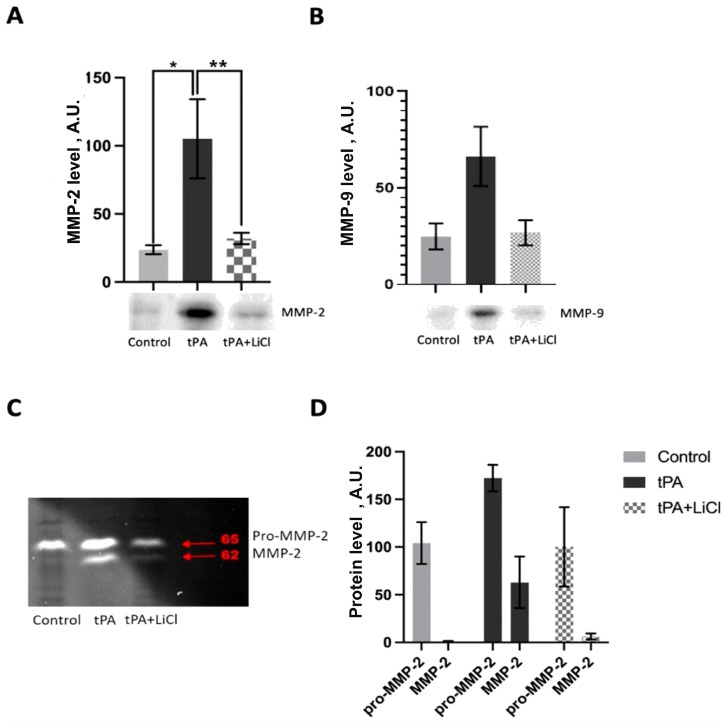
MMP levels in EA.hy926 endotheliocytes treated with 1.6 µM tPA and 1 mM LiCl. The content of MMP-2 (**A**) and MMP-9 (**B**) measured by Western blotting (One-way ANOVA test). Representative images and densitometry analysis of the signal intensity are presented. Representative zymogram for MMP-2 activity (**C**) and its densitometry (**D**) (two-way ANOVA test). The data are presented as mean±SEM. *—*p* ≤ 0.05, **—*p* ≤ 0.005.

**Figure 6 biomedicines-12-01325-f006:**
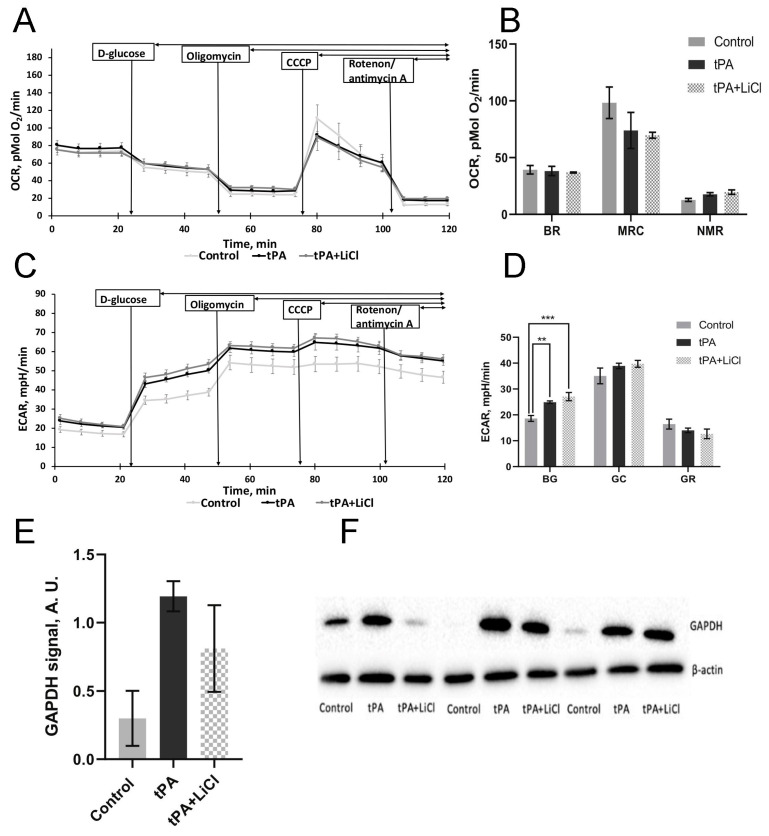
Effect of tPA on glycolysis and respiration of EA.hy926 cells. Effects of tPA and LiCl on oxygen consumption rate (OCR) curves (**A**) and respiratory parameters (**B**): basal respiration (BR), maximal respiration capacity (MRC), and non-mitochondrial respiration (NMR). (**C**) Effects of tPA and LiCl on extracellular acidification rate (ECAR). OCR and ECAR of EA.hy926 (**A**,**C**) were measured after sequential addition of D-glucose (10 mM), ATP synthase inhibitor oligomycin (4.5 μM), carbonyl cyanide m-chlorophenyl hydrazone (CCCP) (10 μM), and a mixture of rotenone and antimycin A (2.5 and 4 μM, respectively). (**D**) Glycolytic parameters: basal glycolysis (BG), glycolytic capacity (GC), and glycolytic reserve (GR). (**E**,**F**) GAPDH content in endothelial cells measured by Western blotting (One-way ANOVA tests). The data are presented as mean ± SEM. **—*p* ≤ 0.005, ***—*p* ≤ 0.001.

## Data Availability

The data that support the findings of this study are available from the corresponding author (plotnikov@belozersky.msu.ru) upon reasonable request.

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
