# Peer review of "Effects of Lithium Ions on tPA-Induced Hemorrhagic Transformation under Stroke"

_biomedicines, 2024, doi:10.3390/biomedicines12061325_

Round 1

Reviewer 1 Report

Comments and Suggestions for Authors

Please provide details of the statistical methods used, including exact post hoc tests.

Clarifies the rationale for the specific doses of LiCl and tPA used.

For in vitro experiments, include the number of repetitions.

If possible, for transparency, please add a table with some datas, such as exact statistics and post-hoc test results.

At the end of the discussion, suggest directions for future research based on your findings.

Add to the discussion the limitations of this study (use of a single cell line, lack of evaluation of long-term effects, incomplete understanding of mechanisms, etc.).

Author Response

Thank you for reviewing our manuscript and for the opportunity to revise our manuscript. We have modified the text to answer all of the reviewer’s recommendations. We have made the appropriate changes to the text and figures. We appreciate the constructive criticism and believe that our manuscript will not elicit further criticism after these revisions.

Below, we present the reviewer's specific comments with our replies.

  • Please provide details of the statistical methods used, including exact post hoc tests.

Answer: We apologize that the statistics are not described correctly.

We have added details about our statistical analysis. Statistical analysis was performed using GraphPad Prizm 7 (GraphPad Software Inc., USA). Data are presented as the mean ± standard error of the mean (SEM). Data were tested for normal distribution using the Shapiro-Wilk test. Comparisons among multiple groups in the in vivo experiment (LPT and EBD data) and in the in vitro experiment (survival of EA.hy926 cells and zymography data) were performed with a two-way ANOVA with repeated measures followed by a post-hoc Sidak test, and other comparisons between multiple groups were performed with a one-way ANOVA followed by a post-hoc Sidak test. A significance level was set at p≤0.05 (*—p≤0.05, **—p≤0.005, ***—Ñ€≤0.001).

  • Clarifies the rationale for the specific doses of LiCl and tPA used.

Answer: For tPA treatment, we used an intravenous injection of tPA at dose 10 mg/kg that is a conventional dose for rats for mimics clinical recanalization (see for instance https://doi.org/10.1038/srep16026).

Rats in the PT+tPA+LiCl group were administered 180 mg/kg LiCl intraperitoneally 1 hour after PT. In previous works, when modeling ischemic stroke by mid cerebral artery occlusion (MCAO) in rats, we used a dose of LiCl 60 mg/kg (Silachev et al, 2015 https://doi.org/10.1016/j.cbi.2015.06.012 ), which had a significant neuroprotective effect. The literature indicates doses of lithium chloride up to 100 mg/kg in different models of brain damage (Boyko et al., 2015 doi: 10.1155/2015/916234). The review of Bartolozzi et al., 2024 (doi: 10.1124/pharmrev.120.000007) indicates that sufficiently high dosages should be used for the therapeutic manifestation of the inhibitory effects of lithium directly on target proteins and subsequent neuroprotection.

In our preliminary experiments with tPA we found some effect of the lithium salt only with 180 mg/kg was the minimum, at which the neuroprotective effect of lithium was observed compared to tPA-treated rats. Thus, it was the lowest effective dose and we used it further.

For in vitro experiments, include the number of repetitions.

Answer: thank you for your comment, we have added the information in the materials and methods. In all in vitro experiments apart from MTT-test the number of replicates was 3. In cell viability assay the number of replicates was 10.

  • If possible, for transparency, please add a table with some datas, such as exact statistics and post-hoc test results.

Answer: We have added more details about statistics and the significance to the figure legends. We have added more details in “Materials and Methods” section (see answer to question 1).

  • At the end of the discussion, suggest directions for future research based on your findings.

Answer: We have added appropriate phrase to the discussion together with the description of limitations of the study.

  • Add to the discussion the limitations of this study (use of a single cell line, lack of evaluation of long-term effects, incomplete understanding of mechanisms, etc.).

Answer: We have added the limitation section to the discussion.

Reviewer 2 Report

Comments and Suggestions for Authors

Comments to the Authors (Minnor comments)

Under conditions of OGD, tPA treatment increased MMP-2/9 levels in endothelial cells, and preincubation with LiCl abolished this 2MMP activation

-How contribute both MMP-9 and 2 produciton by endothelial cells to remodelate the penumbra area and also induce angiogenesis or other events associated to repair after tPA treatment in ischemic animals?

-The stabilization of the blood-brain barrier (BBB) is associated with inhibition of MMP-9 activity [23]. Chronic Li+ treatment for 14 days resulted in neurovascular remodeling via an increase in MMP-9 levels.What does mechanisms can explain the differential acute or subcronic effect of Lithium on MMP-9 activity in terms of repair or damage in cerebral ischemia?

NOTE for Biomedicines Editorial. In case you resend me the maniscript for R1 evaluation, I would like to review the manuscript again in such case if your agree. Recently, I have evaluated other different manuscript for Biomedicines and the R1 version was not visible and and that time the evaluation was Major revision. This is not the case

Material and methods

Experimetal design

Lin 83. The experimental design indicates that an intact group that received no intervention; it it better to say controls (without pharmacoogical intervention) or sham animals in case they receive some kind of manuipulation.

The induced photothrombosis (PT) in the prefrontal cortex followed by administration of  saline group should indicate the duration of rose bengale time for phottrombosis induction. The same with t-PA groups. Please, indicate the dose and duration of t-PA treatment in t-PA -treated animals.Also indicate the Li concentration and duration of the treatment in these Li-trated rats.

Line 100. Why you use the concentration of 180 mg/kg LiCl intraperitoneally 1 hour after PT induction? Please, indicate some published paper that confirm the use of this concentration. The same for the treatment with 40 mg/Kg of rose bengale for PT induction in rats.

-Why was selected 6 hours after PT-induction as intervention time?

Line 126. However, the integrity of BBA has been evaluated by the Evans blue staining at 7th day after the PT following cyte [30] with modifications. However, we selected 3and 6 days after PT induction afor BBA dysruptiion evaluation? I should expect the BBA disruption can occur at 3 days also after PI induction or even earlier? Shall you explain these selected times?

Line 106. Since they evaluated the neurological status of rats on the 3rd and 6th days post-photothrombosis using a limb-placing test (LPT), I was wondering if is enougth to promote repair at this early time after stroke induction. In fact, models of MCAO induction have demonstrated repair mechanisms at 14 or 28 days later. Please, describe repair mechanisms by which Li promote repair in your PT model.

Point 2.5 (line 125-130). They indicate ¨After 30 minutes, the animals underwent intracardiac perfusion with 200 ml (PBS to remove the intravascular Evans Blue¨. It is this time enough to remove Evans blue content?

Line 135.Explain why you use this trichloroacetic acid in 50% ethanol for the homogenation of brain in 3:1 ratio, followed by  centrifugation at 10 000 g for 20 min.

Line 158. Point. 2.7. They indicate that for modeling of ischemic stroke in vitro cells were exposed to oxygen-glucose deprivation (OGD). EA.hy926 were seeded onto 96-well plastic plates 25,000 cells/well in the culture medium, cultured 2 days before OGD. Cell medium was replaced with Dulbecco′s Phosphate Buffered Saline. Then cells were incubated inhypoxic atmosphere with 1% O2 and 99% N2 at 37oC for 4 hours

Shall you explain the chemical composition of the medium for inducting hipoxic environment?

Line 174. They indicate for Western blot and zymography that cells were incubated with 1,6 μM tPA or with 1,6 μM tPA and 1 mM LiCl for 22 hours. However, for the metabolic assay cells were incubated with 1,6 μM tPA for 22 hours.  Why these exactly times? Explain it

Please, shall you explain why you used these doses for t-PA.

Explain why collagecase II sample is used as positive control for MMP Zymograghy

Line 210. Why EA.hy926 were seeded in Seahorse 8-well miniplates at 15,000 cells/well and cultured for  2 days under standard conditions. However, you seeded more cells.

-Why they chose doses of oligomycin (4.5 μM), (CCCP) (10 μM), rotenone and antimycin (2.5 and 4 μM respectively) during 130 min.

Maybe, they don`t find infart volumen reductions (fugure-1) in PL-tPA-Li groups at 7 days after PT induction because this time is not enough for repair the damage cortex. Have you analzyzed other post-PT times in your model?

Line 250. In fact, the absence of effect on volume infarct 7 days after PT induction is not agree with the observed significant reduction at 6 days post-photothrombosis in the PT+tPA+LiCl group (9.5±2.3) compared to the PT (6±0.6) group and PT+tPA (5.3±2.3) group (p=0.0245 and 251 p=0.0075 respectively by two-way ANOVA.Why the infarct volume is unaffected at 6 days after PI-induction?

Line 285. Explain why the tPA had a cytotoxic effect on EA.hy926 cells both alone and in combination with 284 OGD for tPA concentrations (0.4, 0.8, 1.6, 3.2 µM)

Line 289 .Since Pre-incubation with 1 mM LiCl had no significant effect on the cell viability of  EA.hy926 in control groups and after 22-h treatment with 0.8 and 1.6 μM tPA in normoxia (Fig. 4, B), shall you explain why you expect that human endothelial cells can promote protective effects against ODG in these cells?

Please, explain the molecular mechanism by which MMP-2 and MMP-9 promote overactivation by t-PA in vitro and how Lithium prevents this effect.

Line 325. Why lithium chloride treatment failed to influence endothelial cell respiration but treatment with 1,6 μM tPA for 22 hours resulted in elevated ECAR in EA.hy926 cells?.

I would suggest to increase the size of figure 6°A and fig 6 C because it is difficult to see now.

Discussion

Line 372. Haupt et al. (2020) have shown that in mouse middle cerebral artery model occlusion (MCAO) lithium (2 mmol/kg, each at 24 h and at 48 h after  MCAO) stabilizes post-stroke blood-brain barrier via MAPK/ERK/pathway activation, decreases activity and expression of MMP-9 independent of caveolin-1 [23]

Is there a similar effect after PT induction in your model? Is reduced the expression of MMP-9 independent of caveolin-1 after PT induction?

Explain better the possible involvement of MMPs activation in hemorrhagic transformation [33,34] and the contribution of Li on this mechanism in the discussion. How MMPs can be regulated through glycolysis in other cell types? Is there any connextion between MMP-2 or 9 overactivatiion and cell cycle protein dysregulation in your model?

-line 408. How glycolisis can affect MMP9 donwregulated activity whiout affecting MMP2 in human cells line (i.e: human epithelial , carcinoma cells or macrophages in vitro)?

My Decission is minnor revision.

I would suggest to improved these smal changes before the final formal acceptation.

Thanks¡

Comments on the Quality of English Language

It is necceary to improve the english style.

Author Response

Thank you for reviewing our manuscript and for the opportunity to revise our manuscript. We have modified the text to answer all of the reviewer’s recommendations. We have made the appropriate changes to the text and figures. We appreciate the constructive criticism and believe that our manuscript will not elicit further criticism after these revisions.

Below, we present the reviewer's specific comments with our replies.

  • Under conditions of OGD, tPA treatment increased MMP-2/9 levels in endothelial cells, and preincubation with LiCl abolished this 2MMP activation.

-How contribute both MMP-9 and 2 produciton by endothelial cells to remodelate the penumbra area and also induce angiogenesis or other events associated to repair after tPA treatment in ischemic animals?

Answer: The damage to the BBB which manifests in an increase in its permeability, is related to the damage of the vascular endothelium and the destruction of its basal membrane [29,30], which is caused, among other factors, by the increased activity of MMPs, especially MMP-9 and MMP-2. Their proteolytic properties contribute to the destruction of the components of the BBB and increase its permeability. The effect of tPA on the activity of MMPs is apparently one of the main reasons for the increased risk of HT during thrombolytic therapy [31]. (see references in the review: doi: 10.3390/jpm13071175). Unfortunately, there are no studies discussing how MMPs influence the penumbra or induce angiogenesis.

  • -The stabilization of the blood-brain barrier (BBB) is associated with inhibition of MMP-9 activity [23]. Chronic Li+ treatment for 14 days resulted in neurovascular remodeling via an increase in MMP-9 levels. What does mechanisms can explain the differential acute or subcronic effect of Lithium on MMP-9 activity in terms of repair or damage in cerebral ischemia?

Answer: Thank you for your question. So far, most studies on hemorrhagic transformation and the role of MMP in it have not revealed any precise mechanisms. This is one of the reasons why we also conducted this study. The studies we refer to, mainly describe the phenomenon of a decrease or increase in the activity (or number) of various MMPs in stroke or the effect of tPA. For example, the study [24], where an increase in MMP-9 levels by chronic Li+ treatment was detected, does not provide the data about MMP-9 activity (only protein levels). Also, in [24] authors showed the beneficial effects of lithium after the 90-min transient middle cerebral artery occlusion and an increase in MMP-9 levels in the peri-infarct zone by immunohistochemistry. However, there is no discussion of the role or mechanisms of MMP-9 in such chronic intervention with lithium. In particular, we hypothesize that lithium may protect the cells of the neurovascular unit from the cytotoxicity of tPA and thus preserve the integrity of the BBB. Another mechanism lies in the influence of bioenergetics, where changes in the amount of glycolysis intermediates affect the STAT family transcription factor system and thereby regulate the amount of MMPs. This is exactly what we have shown in our work, because we have not only seen a change in the activity of the MMP, but also in its levels (i.e. its expression).

Material and methods

Experimental design

  • Lin 83. The experimental design indicates that an intact group that received no intervention; it it better to say controls (without pharmacological intervention) or sham animals in case they receive some kind of manuipulation.

Answer: We have changed the description and grop’s name according to the reviever’s comments. We have modified the Fig. 2 in the manuscript. Thank you for your advice.

  • The induced photothrombosis (PT) in the prefrontal cortex followed by administration of saline group should indicate the duration of rose bengale time for phottrombosis induction. The same with t-PA groups. Please, indicate the dose and duration of t-PA treatment in t-PA -treated animals.Also indicate the Li concentration and duration of the treatment in these Li-trated rats.

Answer: LiCl and tPA were applied by bolus injection. We have added this to the Methods section as well as other details pf the PT induction.

  • Line 100. Why you use the concentration of 180 mg/kg LiCl intraperitoneally 1 hour after PT induction? Please, indicate some published paper that confirm the use of this concentration. The same for the treatment with 40 mg/Kg of rose bengale for PT induction in rats.

Answer: We used the concentration of 180 mg/kg LiCl intraperitoneally 1 hour after PT induction because it was the minimal dose, at which the neuroprotective effect of lithium was observed in our model of tPA neurotoxicity.

In previous works, we usually used a dose of LiCl about 60 mg/kg in the modeling ischemic stroke by middle cerebral artery occlusion (MCAO) in rats (Silachev et al, 2015 https://doi.org/10.1016/j.cbi.2015.06.012 ), which had a neuroprotective effect. In the literature, doses of lithium chloride of up to 100 mg/kg are reported in various models of brain damage (Boyko et al., 2015 doi: 10.1155/2015/916234). The review by Bartolozzi et al, 2024 (doi: 10.1124/pharmrev.120.000007) indicates that for the therapeutic manifestation of the inhibitory effect of lithium directly on the target proteins and subsequent neuroprotection, sufficiently high doses should be used, up to 1.2 mM. Moreover, conventionally, we and others use lithium as a pretreatment before stroke, so effective dose could be lower, whereas we use LiCl beyond the therapeutic window for stroke (about 3 h post ischemia). In preliminary experiments with tPA, administration of lithium chloride had no protective effect at doses lower than 180 mg/kg, so we began to gradually increase the dose and found that 180 mg/kg was effective.

As for Rose Bengal, researchers are using different doses of rose bengal to modulate photothrombosis in rodents (doi: 10.1016/j.jksus.2023.102838, doi: 10.3791/50370, doi: 10.1186/2040-7378-4-13). Before starting the experiment, we tried to use different dye doses and laser exposure times to select the best conditions for reproducing the model in our laboratory (survival of the animals with the formation of a clear focus of damage and neurological deficit). So, we choose 40 mg/kg and 15 minutes of irradiation.

  • -Why was selected 6 hours after PT-induction as intervention time?

Answer: We wanted to study the most negative effects of tPA thrombolytic therapy in the PT model. In the preliminary experiments we tried to use different timepoints for tPA administration and the most severe adverse effects was found when we used intravenous injection of tPA at 6 hours after stroke modeling.

  • Line 126. However, the integrity of BBA has been evaluated by the Evans blue staining at 7th day after the PT following cyte [30] with modifications. However, we selected 3and 6 days after PT induction afor BBA dysruptiion evaluation? I should expect the BBA disruption can occur at 3 days also after PI induction or even earlier? Shall you explain these selected times?

Answer: For Evans blue staining, animals were given an intravenous injection of 2% Evans Blue dye in saline into the jugular vein. After 30 minutes, the animals underwent intracardiac perfusion with 200 ml of phosphate-buffered saline (PBS) to remove the intravascular Evans Blue dye. The brain was then isolated, photographed and the injured area was removed and weighed. So ,we can only use this method as a terminal point of the experiment, since it requires sacrificing af an animal. The rats cannot be examined for their neurological status, and an MRI cannot be performed after Evans blue-test. Therefore, we first performed the LPT on the 3rd and 6th day and on the 7th day we performed the cylinder test, MRI and only after all tests the Evans blue staining. The test on day 3 would require an additional series of experiments with the additional use of rats. According to the decision of the ethics committee, such an excessive number of animals was considered impractical and only possible if there were no significant effects of tPA and LiCl on day 7. Since we found significant effects of tPA and LiCl on day 7, an additional series of experiments with an analysis of the BBB on day 3 was not performed

  • Line 106. Since they evaluated the neurological status of rats on the 3rd and 6th days post-photothrombosis using a limb-placing test (LPT), I was wondering if is enougth to promote repair at this early time after stroke induction. In fact, models of MCAO induction have demonstrated repair mechanisms at 14 or 28 days later. Please, describe repair mechanisms by which Li promote repair in your PT model.

Answer: the recovery of the neurological state after brain damage in rodents is quite rapid. In some studies, on the mechanisms of lithium treatment in the MCAO model, the promotion of repair was demonstrated as early as after 7 days (doi: 10.1038/s42003-022-03051-2). And we can see the improvement of neurological status on day 6 in our experiment.

In the stroke model studies, lithium (Li+) appears to support recovery through several mechanisms. We found that tPA increased levels of the MMP-2 and MMP-9 in endothelial cells (EA.hy926). These enzymes are known to contribute to hemorrhagic transformation by degrading components of the extracellular matrix and the blood-brain barrier (BBB). It was observed that pretreatment with lithium chloride (LiCl) attenuated the increase in MMP-2 and MMP-9 induced by tPA. This suggests that Li+ may reduce the deleterious effects of tPA on BBB integrity by regulating MMP activity. By reducing MMP activity, Li+ helps to maintain the integrity of the BBB after tPA treatment. This is critical because BBB degradation is a key factor in the pathology of hemorrhagic transformation after ischemic stroke. Although the direct link between MMP activity and glycolysis in endothelial cells has not been extensively studied, there is evidence of a regulatory relationship in other cell types. It is possible that Li+ influences metabolic pathways and stabilizes cellular energy metabolism under ischemic conditions. We have given our arguments about the possible mechanisms of lithium's influence in the discussion section

  • Point 2.5 (line 125-130). They indicate ¨After 30 minutes, the animals underwent intracardiac perfusion with 200 ml (PBS to remove the intravascular Evans Blue¨. It is this time enough to remove Evans blue content?

Answer: 30 minutes is enough for EBD to bind with serum albumin and be evenly distributed throughout all tissues of the body in rats. Then we remove the intravascular Evans Blue dye with PBS perfusion and assess EBD extravasation in the brain. The dye is not removed in 30 minutes, but during perfusion with PBS. The volume of 200 ml corresponds to 10 times the volume of circulating rat blood, so that all the blood is removed from the rat's body together with the circulating EBD.

  • Line 135.Explain why you use this trichloroacetic acid in 50% ethanol for the homogenation of brain in 3:1 ratio, followed by centrifugation at 10 000 g for 20 min.

Answer: Following the method described [30], we found that it is too difficult to homogenize small brain samples in 50% trichloroacetic acid. Therefore, we modified the method: The brain samples were homogenized in PBS, then mixed with solvent (50% trichloroacetic acid: ethanol = 1 : 3) and centrifuged at 10 000 g for 20 minutes. The supernatant of the homogenized brain was collected and the fluorescence of EBD was measured using a microplate reader (ZENYTH 3100, Anthos (Biochrom), UK) at an excitation wavelength of 535 nm and an emission wavelength of 625 nm. Trichloroacetic acid precipitated all proteins from the sample, and ethanol enhanced the fluorescence of EBD.

  • Line 158. Point. 2.7. They indicate that for modeling of ischemic stroke in vitro cells were exposed to oxygen-glucose deprivation (OGD). EA.hy926 were seeded onto 96-well plastic plates 25,000 cells/well in the culture medium, cultured 2 days before OGD. Cell medium was replaced with Dulbecco′s Phosphate Buffered Saline. Then cells were incubated inhypoxic atmosphere with 1% O2 and 99% N2 at 37oC for 4 hours. Shall you explain the chemical composition of the medium for inducting hipoxic environment?

Answer: For OGD modeling we used conventional protocol with commercially available Dulbecco's Phosphate Buffered Saline (BioinnLabs, Russia). The chemical composition of the solution is: 8mM NaHPO4, 137.4 mM NaCl, 1.47 mM KH2PO4, 2.7 mM KCl, 0.9 mM CaCl2, 0.5 mM MgCl2. We have added this information to the Methods section.

  • Line 174. They indicate for Western blot and zymography that cells were incubated with 1,6 μM tPA or with 1,6 μM tPA and 1 mM LiCl for 22 hours. However, for the metabolic assay cells were incubated with 1,6 μM tPA for 22 hours. Why these exactly times? Explain it.

Answer: The standard time used for "one day treatment" is 24 hours. However, prior to tPA treatment there was OGD (4 h), and in some cases we have performed incubation with lithium for 3 hours before OGD. Therefore, the total duration of the preparation phases for incubation and subsequent incubation manipulations is so long that we would not be able to fit in a working day with a 24-hour incubation. Thus, we used 22-h incubation.

  • Please, shall you explain why you used these doses for t-PA.

Answer: for tPA treatment, we used an intravenous injection of tPA at dose 10 mg/kg that is a conventional dose used in rats for mimics recanalization in clinics (see for instance https://doi.org/10.1038/srep16026).

  • Explain why collagecase II sample is used as positive control for MMP Zymograghy

Answer: Collagenase II is a well-known gelatinase that cuts gelatin in the gel. We used collagenase II as the “positive control” only as a marker for sure that all solutions work correctly (we didn’t normalize anything on that).

  • Line 210. Why EA.hy926 were seeded in Seahorse 8-well miniplates at 15,000 cells/well and cultured for 2 days under standard conditions. However, you seeded more cells.

Answer: This is because of the Seahorse well’s size. Seahorse well (in 8-well plate) are smaller than the well from 96-well culture plate. And for in our measurement, we seeded the similar cell count in all experiments.

  • -Why they chose doses of oligomycin (4.5 μM), (CCCP) (10 μM), rotenone and antimycin (2.5 and 4 μM respectively) during 130 min.

Answer: this is recommendations by the Seahorse manufacturer for all such studies.

  • Maybe, they don`t find infart volumen reductions (fugure-1) in PL-tPA-Li groups at 7 days after PT induction because this time is not enough for repair the damage cortex. Have you analyzed other post-PT times in your model?

Answer: We did not expect the lesion to be reduced by the use of LiCl, as it was administered to the animals when the lesion was almost formed (beyond the effective therapeutic window) and it was difficult to reduce its size. tPA, in turn, did not significantly increase the size of the lesion, although the tendency to increase was observed, as we expected. We did not analyze other post-PT times because the end of our experiment (depending on the Evans Blue dye extravasation method) was on day 7. For the study, we used as many animals as possible to obtain the maximum amount of data, which was approved by the ethics committee, but additional timepoint would require more animals.

  • Line 250. In fact, the absence of effect on volume infarct 7 days after PT induction is not agree with the observed significant reduction at 6 days post-photothrombosis in the PT+tPA+LiCl group (9.5±2.3) compared to the PT (6±0.6) group and PT+tPA (5.3±2.3) group (p=0.0245 and 251 p=0.0075 respectively by two-way ANOVA.Why the infarct volume is unaffected at 6 days after PI-induction?

Answer: Infarct volume does not always correlate with neurological status of the animals (doi: 10.1007/s10517-009-0489-z.).

  • Line 285. Explain why the tPA had a cytotoxic effect on EA.hy926 cells both alone and in combination with 284 OGD for tPA concentrations (0.4, 0.8, 1.6, 3.2 µM)

Answer: It has been demonstrated that an increase in tPA concentration from 0 to 3.2 µM results in a gradual decrease in cell survival under both normoxic and OGD conditions, indicating a direct cytotoxic effect of tPA in our study (Fig. 4, A). Our results are consistent with the data by Kenna et al. (2020) who investigated the cellular toxicity of recombinant tPA (rtPA) and examined the effects of rtPA on cell viability in neuronal, astrocyte and brain endothelial cell (bEnd.3) cultures with and without prior exposure to OGD. The study demonstrated that a 4-h or 24-h exposure of rtPA was cytotoxic, affecting neuronal cell viability at ≥ 2 µM, and astrocyte and bEnd.3 cells viability at ≥ 5 μM. In addition, a 4-h exposure to rtPA after a period of OGD exacerbated toxicity, affecting neuronal, astrocyte and bEnd.3 cell viability at rtPA concentrations as low as 0.1 µM (https://doi.org/10.1007/s11064-020-03004-3). It should be noted that in the above-mentioned study, OGD activated MMP-9 release were demonstrated into the supernatant of astrocyte and bEnd.3 cell cultures, but not neuronal cultures, and that OGD/rtPA increases MMP-9 activation (https://doi.org/10.1007/s11064-020-03004-3).

The exact mechanism by which tPA enhances ischemic injury remains to be further investigated. However, one of the possible mechanisms was shown by Gong et al., 2019. They demonstrated for the first time that Sonic hedgehog (Shh) signaling pathway is involved in the tPA-induced reduction of trans-endothelial electrical resistance in brain microvascular endothelial cells (BMECs). Inhibition of the Shh signaling in BMECs might contribute to the tPA-induced disruption of the endothelial barrier and promote cell injury (doi: 10.1007/s11064-018-2697-2).

We have included this discussion in the text of the manuscript.

  • Line 289 .Since Pre-incubation with 1 mM LiCl had no significant effect on the cell viability of EA.hy926 in control groups and after 22-h treatment with 0.8 and 1.6 μM tPA in normoxia (Fig. 4, B), shall you explain why you expect that human endothelial cells can promote protective effects against ODG in these cells?

Answer: It has been shown that lithium salts possess pronounced neuroprotective effects in acute cerebral pathologies [18,19]. Randomized clinical trials on the effect of Li+ in post-stroke patients showed improved motor recovery after early treatment with a low dose of lithium carbonate [20]. A protective effect of lithium ions was observed for endothelial cells, astrocytes [21] and neurons [22] exposed to ischemia. In both oxygen-glucose deprivation (OGD) and mice stroke model Li+ treatment resulted in an increase in the expression of tight junction proteins [23], indicating stabilization of the blood-brain barrier (BBB), which was associated with inhibition of MMP-9 activity [23].

Thus, based on available data, we expected the protective effects against ODG (and/or OGD+tPA) by lithium salts.

  • Please, explain the molecular mechanism by which MMP-2 and MMP-9 promote overactivation by t-PA in vitro and how Lithium prevents this effect.

Answer: The role of MAPK/ERK1/2 signaling pathways in MMP-9 expression, including tPA-induced MMP-9 activation, has been established. The mechanism by which LiCl reduces MMP-9 activity is unclear, but is thought to be related to stimulation of the MAPK/ERK-1/2 signaling pathway. Inhibition of this pathway reversed the Li+-induced decrease in MMP-9 activity and increased BBB permeability. In addition, Li+ treatment upregulates Wnt/β-catenin signaling, thereby preserving tight junctions and endothelial function and decreasing MMP-9 expression. However, the mechanism of the effects of Li+ on BBB dysfunction remains to be elucidated.

Our results showed an increase in glycolysis after tPA treatment, along with higher GAPDH expression. Although the relationship between MMP activity and glycolysis has not been extensively studied, there is evidence that MMPs may be regulated by glycolysis in other cell types. The exact mechanism of the glycolysis-MMP relationship is unclear, but HIF-1-dependent regulation may play a role, as tPA treatment increases HIF-1 expression in the ischemic brain. Since HIF-1 controls the transcription of genes involved in glycolysis and MMPs, it could be a crucial mediator between MMP activity and glycolytic shift in tPA-treated cells.

We have discussed these mechanisms in the manuscript.

  • Line 325. Why lithium chloride treatment failed to influence endothelial cell respiration but treatment with 1,6 μM tPA for 22 hours resulted in elevated ECAR in EA.hy926 cells?.

Answer: We do not provide data on the effect of lithium chloride itself on the endothelium. However, we have performed such experiments and found no pronounced effect of lithium on respiration (OCR) or glycolysis (ECAR). At the same time, tPA treatment increased glycolysis (increased ECAR), which could be related to the observed increase in GAPDH levels. Lithium, in turn, reduced the amount of GAPDH, but this only led to a slight decrease in ECAR. It is likely that the increase in glycolytic activity is associated not only with an increase in the level of this enzyme, but also with other checkpoints that regulate the rate of glycolysis. However, we do not know exactly which mechanisms are responsible for this, as changes in bioenergetics induced by tPA or LiCl have never been studied before

  • I would suggest to increase the size of figure 6°A and fig 6 C because it is difficult to see now.

Answer: thank you, we fixed it. Fig. 6 was modified in the manuscript

Discussion

  • Line 372. Haupt et al. (2020) have shown that in mouse middle cerebral artery model occlusion (MCAO) lithium (2 mmol/kg, each at 24 h and at 48 h after MCAO) stabilizes post-stroke blood-brain barrier via MAPK/ERK/pathway activation, decreases activity and expression of MMP-9 independent of caveolin-1 [23]. Is there a similar effect after PT induction in your model? Is reduced the expression of MMP-9 independent of caveolin-1 after PT induction?

Answer: Thank you for the question. In our study we did not have the opportunity to test this hypothesis. It can be the task for the future investigation.

  • Explain better the possible involvement of MMPs activation in hemorrhagic transformation [33,34] and the contribution of Li on this mechanism in the discussion. How MMPs can be regulated through glycolysis in other cell types? Is there any connection between MMP-2 or 9 overactivation and cell cycle protein dysregulation in your model?

Answer: Thank you for your advice. We apologize for not sufficiently clarifying the involvement of MMPs activation in hemorrhagic transformation in the discussion (we did it in the Introduction). We have added the speculation on these mechanism to the discussion section.

Unfortunately, we did not have the opportunity to test the connection between MMP-2 or 9 overactivation and cell cycle protein dysregulation. It can be the task for the future investigation.

  • -line 408. How glycolysis can affect MMP9 donwregulated activity without affecting MMP2 in human cells line (i.e: human epithelial ,carcinoma cells or macrophages in vitro)?

Answer: Thank you for your question. It is known that the dynamics of MMP-2 and MMP-9 activity reflect their multifunctional role in stroke. It has been shown that their effects largely depend on the stage of pathology development, and that activation of the same MMPs at different stages of stroke can have both negative and regenerative effects (doi: 10.1007/s00018-019-03175-5).

For references see https://doi.org/10.1007/s11010-016-2837-4

The experiments demonstrate that exposure of cancer cells to glycolytic inhibitor at concentration that does not induce ER stress, downregulates the activity and expression of MMP9 without affecting the expression levels and activity of MMP2. Further mechanistic analysis revealed that the regulation of MMP9 was mediated in a SIRT-1 dependent mechanism and did not alter the NFkB signaling pathway. The overall results presented here, therefore suggest that the use of glycolytic inhibitor, 2DG at concentration that do not affect cell viability or induce ER stress can be an effective strategy to control matrix remodeling.

MMP2 is a constitutive enzyme whereas MMP9 is inducible in nature. The expression of MMP9 has been reported to be regulated by NFkB in most of the cancerous conditions. Experiments were therefore, carried out to check if 2DG mediated downregulation of MMP9 is through inhibition of NFkB pathway. The expression levels of NFkB, its nucleo-cytoplasmic distribution and NFkB reporter assay did not show any significant difference in cells treated with 2DG when compared with the untreated controls suggesting that the 2DG-mediated regulation of MMP9 is not dependent on NFkB signaling pathway.